# Immune Checkpoint Inhibitors and Other Immune Therapies in Breast Cancer: A New Paradigm for Prolonged Adjuvant Immunotherapy

**DOI:** 10.3390/biomedicines10102511

**Published:** 2022-10-08

**Authors:** Andrea Nicolini, Paola Ferrari, Angelo Carpi

**Affiliations:** 1Department of Oncology, Transplantations and New Technologies in Medicine, University of Pisa, 56126 Pisa, Italy; 2Unit of Oncology, Department of Medical and Oncological Area, Azienda Ospedaliera-Universitaria Pisana, 56125 Pisa, Italy; 3Department of Clinical and Experimental Medicine, University of Pisa, 56126 Pisa, Italy

**Keywords:** breast cancer, immunotherapy, tumor immunogenicity, monoclonal antibodies against HER2 family receptors, immune checkpoint inhibitors, DNA damage repair inhibitors, hormone therapy, chemotherapy

## Abstract

**Background:** Breast cancer is the most common form of cancer in women worldwide. Advances in the early diagnosis and treatment of cancer in the last decade have progressively decreased the cancer mortality rate, and in recent years, immunotherapy has emerged as a relevant tool against cancer. HER2+ and triple-negative breast cancers (TNBCs) are considered more immunogenic and suitable for this kind of treatment due to the higher rate of tumor-infiltrating lymphocytes (TILs) and programmed death ligand 1 (PD-L1) expression. In TNBC, genetic aberrations further favor immunogenicity due to more neo-antigens in cancer cells. **Methods:** This review summarizes the principal ongoing conventional and investigational immunotherapies in breast cancer. Particularly, immune checkpoint inhibitors (ICIs) and their use alone or combined with DNA damage repair inhibitors (DDRis) are described. Then, the issue on immunotherapy with monoclonal antibodies against HER-2 family receptors is updated. Other investigational immunotherapies include a new schedule based on the interferon beta-interleukin-2 sequence that was given in ER+ metastatic breast cancer patients concomitant with anti-estrogen therapy, which surprisingly showed promising results. **Results:** Based on the scientific literature and our own findings, the current evaluation of tumor immunogenicity and the conventional model of adjuvant chemotherapy (CT) are questioned. **Conclusions:** A novel strategy based on additional prolonged adjuvant immunotherapy combined with hormone therapy or alternated with CT is proposed.

## 1. Introduction

Breast cancer is the most common cancer in women worldwide. The estimated number of new cases in 2020 was 2,260,000, while that for 2025 is predicted to be 2,500,000 [1]. Thus far, due to relevant advances in early diagnosis and treatment of the disease, a 38% decrease in breast cancer mortality has been attained [2]. However, the occurrences of therapy resistance and tumor heterogeneity remain major challenges to avoiding tumor relapse and progression and obtaining definitive success in breast cancer therapy [3]. For more than a decade, a new molecular classification has divided breast cancer into four specific subtypes with different behaviors and responsiveness to therapy. This classification includes luminal A (ER+PR+, HER2/neu negative, and Ki-67 low), luminal B (ER+PR+, HER2/neu negative, and Ki-67 high), HER2+ (ER+ PR+/−, Ki-67 high/low, and HER2+), and triple-negative (ER−PR−HER2−) subtypes. ER+ and/or PR+ endocrine-dependent breast cancer represents up to more than 70% of the molecular subtypes and the incidence increases with older age [4,5]. Endocrine therapy with aromatase inhibitors (Ais), selective ER modulators (SERMs), and selective ER down-regulators (SERDs) is the mainstay for luminal A and B treatment, with luminal A being less aggressive than all other subtypes [5,6]. HER2+ comprehends about 20% of all breast cancers; it is more aggressive than luminal but less aggressive than triple-negative breast cancer (TNBC) [7,8]. HER2 receptor overexpression is a molecular characteristic that accounts for a higher invasiveness and recurrence rate [3,9]. TNBC does not express ER, PR, or HER2 receptors [10] and roughly represents 15–20% of all breast cancers [11,12]. TNBC, which is the subtype with the worst prognosis, has the highest rate of relapses and mortality likely due to the lack of a specific molecular target, and high heterogeneity [8,13,14]. Because of this, neoadjuvant chemotherapy (NACT) is the standard treatment for TNBC [13]. Overall, breast cancer, particularly the ER+ HER2− sub-group, due to the relatively low rate of tumor-infiltrating lymphocytes (TILs) in the tumor microenvironment (TME), is considered scarcely or moderately immunogenic. However, HER2+ and TNBC, both of which are associated with higher TILs, are considered more immunogenic [15]. In particular, the median rate of TIL infiltration reported in TNBC was 20%, while those in HER2+ and HR+HER2− breast cancer were 16% and 6%, respectively [15]. Further, HER2 overexpression and, more recently, an uncovered higher programmed death ligand 1 (PD-L1) expression in HER2+ and TNBC [16], along with the occurrence of germline breast cancer (gBRCA1/2) genetic defects and homologous recombination deficiency (HRD) in the latter, suggest both are suitable for an immunotherapeutic strategy. This review examines the major ongoing conventional and investigational immune therapies in breast cancer. Particularly, the immune checkpoint inhibitors (ICIs) and their use, alone or combined with DNA damage repair inhibitors (DDRis), which is under investigation for breast cancer, are described first. Then, conventional passive immunotherapy with monoclonal antibodies against HER-2 family receptors is updated. Other investigational immune therapies, including a new immunotherapy schedule successfully tested in a clinical trial carried out in ER+ metastatic breast cancer patients, are briefly reported. Furthermore, based on the scientific literature and our findings, the current evaluation of tumor immunogenicity and the conventional model of adjuvant chemotherapy (CT) are questioned. Finally, a novel strategy based on additional prolonged adjuvant immunotherapy combined with conventional hormones or CT is proposed.

### 1.1. Conventional Immune Therapies in Breast Cancer

The immune checkpoint blockade (ICB) by ICIs is an emerging active immunotherapy for TNBC, while in the HER+ breast cancer subset, passive immunotherapy with monoclonal antibodies against HER-2 family receptors has been carried out for a long time. 

#### 1.1.1. ICIs and ICB in Breast Cancer Therapy

Immune checkpoints are one of the mechanisms by which the immune system promotes immune cells anergy or apoptosis to maintain the immune response homeostasis [16,17]. Immune checkpoints are inhibitory proteins/receptors usually present on the surface of effector immune cells, or T or NK cells, and their ligands are transmembrane proteins expressed on antigen-presenting cells (APCs), macrophages, or tumor cells, as well as B and T lymphocytes; the ligands are also expressed in some nonlymphoid tissues [18]. Cells can either start an immune response through signals or maintain immune tolerance; namely, the binding of the PD-L1 with its programmed death-1 (PD-1) receptor provides a suppressive signal to T lymphocytes with decreased proliferation and a decrease in the immune response [19,20]. T-cell activation is controlled by further antigen-independent co-stimulatory signals such as cluster of differentiation 28 (CD28) and cytotoxic T-lymphocyte-associated antigen 4 (CTLA-4). CD28 and CTLA-4 compete for the same ligands: CD80 and CD86 [21]. However, CTLA-4 has a higher avidity than CD28 with regard to these ligands and induces a negative regulatory signal to the T cell [21]. Particularly and contrary to CD28 signals, which are required for T-cell activation and cytokine secretion, CTLA-4 signaling inhibits T-cell activation. Both CD28 and CTLA-4 can be stimulated by the CD80 (B7-1) and CD86 (B7-2) ligands that are expressed on activated APCs, leading either to T-cell proliferation and differentiation when the CD28:CD80/CD86 ratio is elevated [22], or to T-cell inactivation and anergy in the case of an increased CTLA-4:CD80/CD86 ratio [22]. As CTLA-4 binds to CD80/86 with very high affinity, this receptor mediates immune suppression by competing for CD28 as well as by inducing CD80/86 removal from the APCs’ surface [23]. For this reason, by blocking the interaction between CTLA-4 and CD80/86 ligands, CTLA-4 inhibitors can prevent T-cells exhaustion and boost the antitumor T-cell response [24]. PD-1, CTLA-4, and CD28 receptors are present on activated effector T cells that interact with their ligands, the B7 family members: B7-1 (CD80), B7-2 (CD-86), B7-H1 or CD274 (PD-L1), B7-H2, B7-H3, B7-DC or CD273 (PD-L2), and B7-H4 [21,25]. PD-1 interrupts the effector T-cell responses by interacting with its ligands PD-L1 and PD-L2 [26]. PD-L2 has a much higher affinity for PD-1 than PD-L1 [27], and the PD-L1 blocking antibodies do not affect PD-L2/PD-1 interaction [28]. However, PD-L1 is considered the principal PD-1 ligand. LAG3-MHC class II/Lectins, TIGIT-CD155/CD112, TIM3-Galectin 9/PtdSer/HMGB1, and VISTA/VSIG 3 are other well-investigated receptor–ligand pairs. Overexpression of PD-L1 and CTLA-4 has been reported in HER-2+ and TNBC [13,29]. ICB by ICIs is an emerging therapeutic strategy that aims to restore antitumor immune response and eradicate cancer cells by inhibiting the negative regulators of effector T-cells. Several clinical trials have been developed or are ongoing with some preliminary promising results [30]. 

#### 1.1.2. PD-L1 Expression

In addition to the well-known ER positivity and HER-2-expression/amplification, PD-L1 is another biological molecule that has recently emerged as a potential biomarker in breast cancer [31,32]. Commonly, PD-L1 expression is measured semi-quantitatively by immune histochemistry (IHC) [33] in tissue microarrays or a whole section. Table 1 shows the main parameters and criteria used for PD-L1 expression assessment. Usually, PD-L1 expression is absent in normal breast tissue, while in breast cancer, PD-L1 expression is measured in tumor cells (TCs) and/or immune cells (iCs). Thus far, in a few studies, PD-L1 expression in TCs and ICs has been evaluated with a discordance rate of up to 40–50%, according to the utilized IHC assay [34,35]. IHC PD-L1 expression in breast cancer is lower (10–30%) than in other tumors, such as non-small-cell lung cancer (about 70%) [36], and different based on the molecular subtype. Particularly, PD-L1 is higher in cases of TNBC, followed by the HER2+ subtype, and decreases further in HR+ advanced breast cancer. Moreover, receptor conversion during progression from primary to metastatic disease is well known even for PD1/PD-L1 expression [37,38,39]. This highlights the need to carry out metastatic biopsies as these suggest changing management roughly in 15–20% of patients [39,40,41]. In different metastatic sites, PD-L1 positivity at biopsy, similar to HR and HER-2 expression, varied from 12% to 60% [39]. PD-L1 over-expression occurs in 9–45% in HR+ early breast cancer patients, namely around 9% in luminal A and about 42% in luminal B, while in the metastatic stage, it decreases to 0–1% in luminal A and 10–12% in luminal B [42]. In nonmetastatic HER2+ breast cancer, the PD-L1 positivity rate is 30–35% and 35–60% in early TNBC compared with 9–15% in HER2+ advanced breast cancer and 30–40% in metastatic TNBC [43]. In a recent systematic review and meta-analysis on the PD-L1 status in breast cancer [44], the positivity rate of pooled PD-L1 was increased in primary tumors compared to metastasis when evaluated in IC (51.2% vs. 37.1%, *p* < 0.001) and TC/IC (30.1% vs. 14.6%, *p* < 0.001), unlike in TC (18.7% vs. 17.8%, *p* = 0.65). Moreover, the lowest PD-L1 positivity was observed in bone metastases (12%) and the highest in lymph nodes (60%). Thus, more commonly, the direction of change was from PD-L1-positive primary tumor to PD-L1-negative metastasis than vice versa, independent of the evaluated cell type. These discordances suggest both spatial and temporal heterogeneities during tumor progression, which implies temporal evolution of the immune surveillance at the TME. In particular, the transition from an initial inflamed although immunosuppressed TME to an immune-desert TME [45] is supported by the decrease in TILs at the metastatic sites, as well as a higher PD-L1 positivity concordance in synchronous samples compared to metachronous samples [46].

#### 1.1.3. PD-L1 Assay

Currently, there are multiple PD-L1 IHC assays available with various scoring algorithms that have received approval for different therapies and tumor indications [47]. However, there is no standardized method for PD-L1 assessment. SP142, SP263, 28-82, 22C3, E1L3N, 73–10, E1J2J, 5H1, 4059, and 9A11 are some assays among others [43]; of these, the PD-L1 IHC 22C3pharmDx Assay (PD-L1 22C3, DAKO) and SP142 Assay (PD-L1 SP142, VENTANA) are used most often [47,48,49,50]. Commonly, the approval of immunotherapy drugs joins with a predetermined IHC assay. For example, PD-L1 IC score (PD-L1 SP142, VENTANA) was used for the approval of atezolizumab, whereas PD-L1 combined positive score (with 22C3 assay) is predictive for pembrolizumab. This procedure defines a companion diagnostic assay, which is mandatory to recruit a candidate for treatment with the approved drug. A complementary diagnostic test, unlike a companion diagnostic assay, is not limited to a specific drug and can be used more widely with a class of agents [51]. A tumor proportion score ≥ 1% or a combined positive score ≥ 1 are the criteria most commonly used for indicating PD-L1-positive expression [47] (Table 1). In the Impassion130 trial [52,53] conducted in TNBC patients, a post hoc analysis compared SP263, 22C3, and SP142 assays and found IC ≥ 1% for SP263 and SP142 and a combined positive score of ≥1 for 22C3-defined PD-L1 positivity. Again, there was no concordance of SP263 and 22C3 vs. SP142 (75% and 81% vs. 46%, respectively). Consistent with a mathematical model, a combined positive score ≥10 for 22C3 and an IC ≥ 4% for SP263 were the optimal cut-offs for PD-L1 positivity. However, based on these optimal cutoffs, a low concordance of SP263 and 22C3 versus SP142 was found [53]. Collectively, this suggests that PD-L1 IHC assays are not interchangeable in the current clinical practice.

**Table 1 biomedicines-10-02511-t001:** Main parameters used for the assessment of PD-L1 expression.

Parameter	Evaluation		Ref.
	Criterion	Analysis	
Immune cells (iCs)	%	Percentage of tumor area involved by any intensity of PD-L1 IHC staining	[54]
	Score	(Area of tumor infiltrated by PD-L1+ iCs/total tumor area) × 100%	[33,54,55]
Tumor cells (TCs)	%	(Number of PD-L1+ iCs/total number of viable TCs) × 100%	[33,55,56]
	Tumor cell score	The percentage of tumor area involved by PD-L1+ TCs related to the entire tumor area	[33,56]
	Tumor proportion score	(Number of viable PD-L1+ TCs/total number of viable TCs) × 100%	[33,55,57]
iCs and TCs	Combined positive score	* (Number of PD-L1+ TCs plus iCs/total number of viable TCs) × 100%	[33,55,56]

IHC: immunohistochemistry; * the number of PD-L1+ TCs and iCs is summarized with reference to all viable TCs and thereafter multiplied × 100, while 100 is the highest combined positive score.

#### 1.1.4. Pembrolizumab and Nivolumab (PD-1 Inhibitors) (Table 2)

Pembrolizumab and nivolumab are IgG4 antibodies against PD-1 receptors. The safety and antitumor activity of pembrolizumab monotherapy have been assessed in the phase Ib KEYNOTE-028 trial enrolling heavily pre-treated ER+ HER2-advanced breast cancer patients with PD-L1-positive tumors (combined positive score ≥ 1) who received pembrolizumab monotherapy for up to 2 years or until confirmed progression/intolerable toxicity; the overall response rate (ORR) was 12%, 16% of patients had stable disease (SD), the clinical benefit rate was 20%, and the median duration of response was 12 months. The authors concluded that in the studied population, pembrolizumab was well tolerated with a modest but durable overall response [58]. In the phase II Keynote-086 trial, pembrolizumab monotherapy for pretreated or nonpretreated metastatic TNBC patients was well tolerated and showed prolonged anti-tumor activity [59,60]. These preliminary promising results were not confirmed by the phase III Keynote-119 trial comparing pembrolizumab monotherapy to standard CT in metastatic TNBC patients. In this trial, pembrolizumab monotherapy did not significantly prolong overall survival (OS) [61]. Therefore, pembrolizumab combined with CT versus placebo were tested in the phase III Keynote-355 [62] and Keynote-522 [63] trials. Following the results of these trials [64], the FDA recently approved pembrolizumab in combination with CT for patients with untreated recurrent inoperable or metastatic TNBC and as NACT for patients with untreated early-stage TNBC. In the phase 1b-2 PANACEA Keynote-014 trial, pembrolizumab in combination with trastuzumab was evaluated in trastuzumab-resistant, advanced, and HER2+ breast cancer patients. Among the 52 patients recruited in phase II, 40 had PD-L1+ and 12 had PD-L1− tumors. Six (15%) of 40 PD-L1+ patients achieved an objective response (OR), while there were no objective responders among the PD-L1− patients. The authors concluded that pembrolizumab plus trastuzumab was safe and showed activity and durable clinical benefit in patients with PD-L1+, trastuzumab-resistant, advanced, and HER2+ breast cancer [65]. Some other investigational studies are being conducted with pembrolizumab in different settings and in combination with different drugs (NCT02768701, NCT03121352, NCT03025035, NCT03032107, NCT04251169, NCT03393845, and NCT02752685). In the TONIC trial conducted in metastatic TNBC, nivolumab alone with or without induction therapy showed a 20% ORR [66]. In a single-arm phase II study, the efficacy and safety of cabozantinib combined with nivolumab in metastatic TNBC patients were evaluated. Among the 17 evaluable patients, one patient confirmed partial response (PR), 14 SD, and 2 progressive disease (PD). As the study did not meet the pre-specified criteria, it was closed early. Toxicity led to cabozantinib dose reduction in 50% of patients. The responding patient had a PD-L1-negative tumor with low tumor mutational burden but high TILs and enriched immune gene expression; further, immunostaining, genomic, and proteomic studies indicated a high degree of tumor immune suppression in the studied population [12]. In a prospective, single-arm, open-label, phase 1b trial (NCT03807765), nivolumab and stereotactic radio-surgery were assessed among metastatic breast cancer patients with brain metastases. The initial dose of nivolumab was followed, 1 week later, by stereotactic radio-surgery. A total of 12 patients were treated up to 17 lesions. No dose-limiting toxicities were observed and the most common neurological adverse events (aEs) included grade 1 to 2 headaches and dizziness occurring in 5 (42%) patients. The median intracranial control was 6.2 months with 6- and 12-month control rates of 55% and 22%, respectively [67]. 

#### 1.1.5. Atezolizumab, Durvalumab, and Avelumab (PD-L1 Inhibitors)

Atezolizumab, durvalumab, and avelumab are IgG1 monoclonal antibodies (mAb) and target PD-L1. Avelumab additionally has the capability to induce antibody-dependent cellular cytotoxicity, which is an indirect antitumor effect obtained by engaging mAb with Fc receptors on immune effector cells [68]. Good tolerability and efficacy of atezolizumab were found in a phase I study conducted in metastatic TNBC patients [69]. The Impassion 130 trial was conducted using atezolizumab plus nab-paclitaxel (A+nP) versus placebo plus nab-paclitaxel (P+nP) in patients with locally advanced or metastatic TNBC [70,71,72]. Immune-mediated aEs of special interest were observed in 58% and 41.6% of the patients treated with A+nP and P+nP, respectively [73]. Following the promising results of both trials, the Ventana PD-L1/SP142 IHC assay received the FDA’s approval as an atezolizumab companion diagnostic assay [29]. In the phase III Impassion 031 trial carried out in early-stage TNBC patients, atezolizumab combined with nab-paclitaxel and anthracycline-based CT compared to placebo showed a significantly higher rate of pathological complete response (pCR) with acceptable tolerability [74]. Unlike this, the Impassion 131 phase III trial of first-line paclitaxel with or without atezolizumab for unresectable locally advanced or metastatic TNBC showed that atezolizumab combined with paclitaxel did not increase progression-free survival (PFS) or OS compared to paclitaxel alone [75]. In a phase I study carried out in recurrent cancer patients including TNBC, durvalumab showed acceptable safety and preliminary antitumor activity [76]. In the phase II GeparNuevo study that recruited early TNBC patients, durvalumab was administered as NACT combined with an anthracycline-taxane-based treatment. An increased pCR occurred mainly in patients who had received durvalumab monotherapy before CT [77]. In the phase II SAFIR02-Breast Immuno trial, durvalumab monotherapy improved the outcomes of TNBC patients, compared to maintenance CT [78]. Further, avelumab was tested in a phase I study enrolling locally advanced or metastatic breast cancer patients, including those with TNBC. In this study, an acceptable toxicity and clinical anti-tumoral activity was reported mainly in tumors with PD-L1 over-expression [79]. 

#### 1.1.6. Efficacy and Safety of Anti-PD1/PD-L1 Monotherapy in Metastatic Breast Cancer

Recently, interim analyses to evaluate the efficacy and safety of anti-PD-1/PD-L1 monotherapy for metastatic breast cancer have been carried out; 586 advanced breast cancer patients treated with anti-PD-1/PD-L1 monotherapy agents were included from six studies. The anti-PD-1/PD-L1 agents used for the treatment were pembrolizumab, atezolizumab, and avelumab. Overall, regarding monotherapy, the complete response (CR) rate was 1.26%, the partial response (PR) rate was 7.65%, the ORR was 9.85%, and the disease control rate was 18.33%. The one-year OS rate and 6-month PFS rate were 43.34% and 17.24%, respectively. The incidence of aEs was 64.18% in any grade and 12.94% in severe grade, while the incidence of immune-related (ir)aEs was approximately 14.75%. Moreover, comparing PD-L1-positive and PD-L1-negative groups, the correspondence between efficacy and expression of the PD-L1 biomarker was found as follows: PR was 9.93% vs. 2.69%; ORR was 10.62% vs. 3.07%; disease control rate was 17.95% vs. 4.71%, respectively [80].

#### 1.1.7. Tremelimumab and Ipilimumab (CTLA-4 Inhibitors)

Tremelimumab and ipilimumab are IgG2 and IgG1 mAb, respectively, and are used against CTLA-4. In a phase I dose escalation study, the safety of tremelimumab, administered on the third day of palliative RT, was evaluated in patients with metastatic breast cancer. Among the six enrolled patients, five had HR+ metastatic breast cancer and one had metastatic TNBC. One dose-limiting toxicity occurred at 6 mg/kg; however, the trial closed before MTD could be determined. One patient discontinued treatment due to a pathological fracture. SD was the best response, and one patient had SD for >6 months. The median OS was 50.8 months, with one patient surviving >8 years. Peripheral blood mononuclear cell profiles showed increasing proliferating (Ki67+) Treg cells 1 week post-treatment in five patients. The authors concluded that this combination approach needed to be optimized [81]. Another investigational study with tremelimumab in addition to durvalumab in HR+/HER2− stage I to III breast cancer patients without any previous therapy is ongoing (pilot study NCT03132467). In the ICON, a phase Ib study, immunogenic CT will be evaluated with ipilimumab and nivolumab in metastatic HR+ breast cancer patients. In particular, anthracycline, which is considered “immunogenic”, and low-dose cyclophosphamide, which has been reported to counter immunosuppressive cells, are the combined CT. The trial will enroll 75 evaluable subjects randomized into two arms (A:B). Patients in Arm A will receive only CT, i.e., pegylated liposomal doxorubicin and low-dose cyclophosphamide. Patients in Arm B will receive pegylated liposomal doxorubicin  +  cyclophosphamide  +  ipilimumab + nivolumab. Patients in Arm A will be offered ipilimumab  +  nivolumab after PD. The rationale consists in expecting that CT synergize with ICIs and in combining PD-1 and CTLA-4 blockade, as these checkpoints are important in different phases of the immune response [82]. Table 2 shows the main clinical trials carried out with ICIs in breast cancer and their principal results.

**Table 2 biomedicines-10-02511-t002:** Main clinical trials conducted with immune checkpoint inhibitors (ICIs) in breast cancer. (**A**) Clinical trials with pembrolizumab (PE) or nivolumab (NI). (**B**) Clinical trials with atezolizumab (ATZ), durvalumab (DRV), avelumab (AV), tremelimumab (TREMB) or ipilimumab (IPI).

A								
Trial	Phase	Drug	Mechanism of Action	Setting	Therapeutic Regimen	Pts (n)	Outcome	Ref.
Keynote-028 NCT 02054806	Ib	Pembrolizumab (PE)	PD1 inhibitor	ABC PD-L1+	Monotherapy for up to 2 years	25	ORR 20% SD 16% CB 20% MDR 12 mo.	[58]
Keynote-086 NCT 02447003	II			mTNBC	Monotherapy, >1 prior systemic therapy	170	ORR 5.7% in PDL1+ pts PFS 2 mo. OS 9 mo.	[59]
	Cohort B				Monotherapy, no prior systemic therapy	84	ORR 20% PFS 21 mo. OS 18 mo.	[60]
Keynote-119 NCT 02555657	III			mTNBC	PE vs. CT, prior systemic therapy	312 vs. 310	OS 12.7 vs. 11.6 mo. in PDL1+ pts	[61]
Keynote-355 NCT 02819518	III			mTNBC	PE + CT vs. PB + CT, no prior CT	566 vs. 281	PFS 9.7 vs. 5.6 mo. in PDL1+ pts; PFS 7.6 vs. 5.6 mo.	[62]
Keynote-522 NCT 03036488	III			Early, stage II-III TNBC	PE + PTX + CBD vs. PB + PTX + CBD	401 vs. 201	pCR 64.8% vs. 51.2%	[63]
Panacea-Keynote-014	Ib-II			HER2+, Trastuzumab resistant ABC	PE + trastuzumab	52	ORR 15% in PDL1+ pts No response in PDL1− pts	[65]
TONIC	II	Nivolumab (NI)		mTNBC	(1) NI without induction; (2) NI with induction *; (3) RT; (4) CY; (5) CIS; (6) DOXO 3-4-5-6: followed by NI	67	ORR 20% for all pts CIS 23% DOXO 35%	[66]
Institutional (open-label, single arm, single-center, closed early)	II			mTNBC	NI + cabozantinib	18	ORR 6% (1 PR in a PDL1− pt)	[12]
NCT03807765	IB			BC with brain metastases	NI followed by SRS	12	6 mo. control rate 55% 12 mo. control rate 22%	[67]
**B**								
**Trial**	**Phase**	**Drug**	**Mechanism of Action**	**Setting**	**Therapeutic Regimen**	**Pts (n)**	**Outcome**	**Ref.**
NCT01375842	I	Atezolizumab (ATZ)	PD-L1 inhibitor	mTNBC unselected for PDL1	Monotherapy	116	ORR 10–12% in PDL1+ Median PFS 1.4 mo. Median OS 8.9 mo.	[69]
iMpassion 130 NCT02425891	III			mTNBC unselected for PDL1	ATZ + Nab-PTX vs. PB + Nab-PTX	451 vs. 451	ORR 56–58.9% in PDL1+ Median PFS 7.5 mo. in PDL1+ Median OS 25 mo in PDL1+	[70,73]
iMpassion 031 NCT03197935	III			Stage II or III TNBC, no prior systemic therapy	ATZ + CT vs. PB + CT	165 vs. 168	pCR 58% vs. 41% *p* = 0.0044	[74]
iMpassion 131 NCT03125902	III			Locally advanced or mTNBC, no prior systemic therapy	ATZ + PTX vs. PB + PTX	431 vs. 220	Median PFS 6 vs. 5.7 mo. Median OS 22.1 vs. 28.3 mo in PDL1+ pts	[75]
NCT02484404	I	Durvalumab (DRV)		Recurrent cancers including TNBC	DRV + cediranib + olaparib	9 (1 TNBC)	CBR 67% PR 44%	[76]
GeparNuevo NCT02685059	II			NACT in early TNBC	DRV + Nab-PTX vs. PB + Nab-PTX followed by EC **	174 (117 the window cohort)	pCR 53.4% vs. 44.2% pCR 61% vs. 41.4% in window cohort	[77]
SAFIR 02 Breast Immuno NCT02299999	II			HER2− mBC with prior CT	DRV vs. maintenance CT	32 vs. 29 assessed	HR of death: 0.37 for PDL1+ pts vs. 0.49 PDL1− pts	[78]
Javelin NCT01772004	Ib	Avelumab (AV)		Locally advanced or mTNBC, refractory or progressing	Monotherapy	58 TNBC	ORR 22.2% in PDL1+ pts vs. 2.6% in PDL1− pts	[79]
Institutional (open-label, single-arm, single-center, closed early)	I	Tremelimumab (TREMB)	CTLA-4 inhibitor	Incurable mBC	TREMB + RT	6 (1 TNBC)	Median OS 50.8 mo.	[81]
ICON	Ib	Ipilimumab (IPI)		HR+ mBC	IPI + NI + PLD + CY vs. PLD + CY	75	Expected	[82]

Pts: patients; ABC: advanced breast cancer; mBC: metastatic breast cancer; ORR: overall response rate; SD: stable disease; CB: clinical benefit; CBR: clinical benefit rate; MDR: median duration of response; PFS: progression-free survival; OS: overall survival; mTNBC: metastatic TNBC; CT: chemotherapy; PB: placebo; PTX: paclitaxel; CBD: carboplatin; pCR: pathological complete response; *: 2 weeks low dose; RT: radiotherapy; CY: cyclophosphamide; CIS: cisplatin; DOXO: doxorubicin; PR: partial response; SRS: stereotactic radiosurgery; **: in the window-phase, DRV/PB alone was given 2 weeks before the start of Nab-PTX; PLD: pegylated liposomal doxorubicin.

#### 1.1.8. PD-L1 Testing as Prognostic and/or Predictive Biomarker in Breast Cancer

The clinical usefulness of the PD-L1 assay largely varies between cancer types and different settings of treatment [55]. The Impassion130 trial, an exploratory analysis carried out in PD-L1-iCs-positive patients, showed that the median OS was 25.4 months versus 17.9 months in patients treated with atezolizumab plus nab-paclitaxel (A + nP) vs. nab-paclitaxel plus placebo (nP + P), respectively. In the PD-L1-iCs-negative population, the median OS was 19.7 months in each treatment arm [73]. In a sub-study from the same trial [46], the PD-L1 assay, together with other immune markers, was evaluated for association with clinical benefit. PD-L1 IC > 1% of the tumor area in both primary and metastatic tumors predicted a favorable outcome and joined with PFS benefit with A + nP versus P + nP. An investigational study [83] was carried out to better understand the outcome discrepancy between the Impassion 130 and Impassion 131 clinical trials. In this study, the changes in iCs over time in 22 patients with metastatic TNBC receiving paclitaxel or paclitaxel plus atezolizumab were evaluated. It was found that high levels of baseline CD8-CXCL13 and CD4-CXCL13 could predict an efficacious response and that following treatment with paclitaxel plus atezolizumab versus paclitaxel alone, both CD8-CXCL13 and CD4-CXCL13 expanded in responsive patients after treatment. The authors concluded that the reduction in both these populations by paclitaxel may compromise the clinical outcome of the accompanying atezolizumab. In a study that examined six OS curves from three randomized clinical trials (Keynote-355, Impassion130, and iMpassion131 studies) conducted in metastatic TNBC patients, the efficacy of first-line ICIs combined with CT was analyzed. PD-L1 iCs > 1% positivity for atezolizumab and a PD-L1 combined positive score > 10% for pembrolizumab showed prognostic value. In particular, in patients with PD-L1 iCs > 1% positivity, atezolizumab combined with a significantly better OS than pembrolizumab. As for pembrolizumab, a threshold of PD-L1 combined positive score > 10% increased the median OS from 15.5 to 23 months [84]. In the study by Dirik et al. [79], where 168 heavily pretreated patients with locally advanced or MBC, including 58 patients with TNBC, received avelumab, a trend toward a higher ORR was seen in patients with PD-L1+ versus PD-L1− tumor-associated iCs in the overall population (16.7% vs. 1.6%, respectively), as well as in the TNBC subgroup (22.2% vs. 2.6%, respectively). In the meta-analysis by Qi et al. [80] considering six studies of advanced breast cancer including TNBC and HR+ HER2− patients who had received anti-PD-1/PD-L1 monotherapy, the response rate as reported was closely associated with the expression of the PD-L1 biomarker (PD-L1+ vs. PD-L1−). The PD-L1-positive expression status was defined as PD-L1 expressed in ≥1% TCs or/and tumor-associated iCs. Unlike these favorable results, in the Keynote 086 and Keynote 150 studies recruiting metastatic TNBC patients, pembrolizumab given alone or with eribulin, respectively, did not show significant clinical differences, according to PD-L1 status. Moreover, in the Keynote 086 study, 60% of patients suffered from treatment-related aEs including those that were immune-mediated [11]. Lastly, in the study where nivolumab was given along with cabozantinib for metastatic TNBC [12], a PD-L1-positive tumor occurred in one among 15 patients who were successfully tested; this patient had SD as the best response.

### 1.2. ICIs Combined with DDRi

ORRs for ICB monotherapy in breast cancer range from 5% to 23% [85]. To increase its efficacy, a combinatorial therapy of ICIs with DDRi is particularly promising.

#### 1.2.1. DNA Damage Repair Defects, Tumor Mutational Burden, and Neo-Antigens

The principal task of DNA damage repair (DDR) proteins following DNA damage due to endogenous factors or exogenous insults is to protect the integrity of the genome [86]. Patients with tumors carrying a DNA mismatch repair deficiency (MMRd), HRD, BRCA1/2 genetic defects, or other defects in DDR genes [87] have a higher tumor mutation burden and more neo-antigens. The deficiency of DDR genes is reported in a wide range of malignancies including TNBC [88]. Genomic loss of heterozygosity, large-scale translocations, and telomeric allelic imbalance together with BRCA mutations [89] have been proposed as molecular hallmarks of HRD [90]. HRD occurs in more than 20% of breast cancers and 69% of TNBC [91]. Breast cancers with gBRCA1/2 mutations only account for 5–7% of breast cancer cases [92] mainly in TNBC [93], while a high mutation rate is observed in high-grade TNBC [94]. Overall, tumor mutational burden is a surrogate marker for tumor neo-antigen load and patients with higher tumor mutational burden can likely benefit from DDRi [88], which, through the increase in tumor mutational burden and neo-antigens, can enhance antitumor immunity.

#### 1.2.2. Barriers to ICIs Response and Mechanisms by Which DDRi Synergizes with ICIs

It is commonly thought that the limited ICB efficacy is due to the presence of multi-factorial barriers in the TME, some of which involve DDR signaling pathways that could be reversed by DDRi. The mutation of genes involved in DNA replication pathways is reported to be enriched in patients showing durable clinical benefit from ICB. Considering this, the use of poly adenosine diphosphate-ribose polymerase inhibitors (PARPis) or other DDRi-targeting DDR pathways can potentially improve ICB for cancer therapy. Low tumor mutational burden, particularly low true neo-antigen burden, which is the number of alterations actually addressed by effector T-cells, is one barrier that has a strong impact on ICB response. Tumors with high microsatellite instability acquire many somatic mutations due to the failure of DNA MMRd. The following high tumor mutational burden makes tumors more immunogenic and sensitive to PD-1 ICB. Mutant neo-antigens in MMRd cancers make them sensitive to ICB therapy in many different tumor types [95,96], suggesting a pan-tumor biomarker function for ICB efficiency. However, MMRd was only observed in around 1% of breast cancers, with 1.8% in TNBC [97]. HRD has less impact than MMRd on tumor mutational burden; however, the genomic instability in HRD tumors can favor DDRi responsiveness when given together with ICB [98]. A low PD-L1 expression can be another barrier to ICB response. DNA damage enhanced by DDRi can increase PD-L1 expression by activating both IRF1 and interferons [99]. The adaptive PD-L1 up-regulation, which counteracts the PARPi-mediated immune activation, can likely be overcome by the combinatorial treatment of ICB with DDRi [98]. ICB can also be ineffective due to a dysfunctional T-cell phenotype, as it occurs with T-cell apoptosis, which is a mechanism of tumor-induced T-cell dysfunction triggered by the binding of different molecules on T-cells with the corresponding ligands [100]. Impaired DNA machinery favors T-cells apoptosis by the inhibition of telomeric repeat binding factor 2, telomerase, topoisomerase I and II alpha, and ATM kinase actions [101], while amplifying DNA damage by DDRi can reprogram TME. This reprogramming induces the recruitment of T-cells into the tumor bed [102], thus transforming “cold tumors” into “hot tumors” through the activation of the immune response. Down-regulated tumor MHC-I/II expression has been reported to contribute to ICB resistance [103]. On the other side, DNA damage favored by DDRi can increase MHC-I molecules expression on the surface of tumor cells [104], thus supporting ICB reversal. Experimental findings show that drugs targeting proteins involved in the DDR pathway, apart from PARPi, significantly synergize with anti-PD-L1 therapy, suggesting that DDRi can successfully impact immunosuppressive cells [98]. Altered genes associated with different pathways including MAPK, JAK-STAT, PI3K-AKT, WNT-beta catenin, and Hippo have been reported to affect the ICB response. Moreover, two studies found that STK11/LKB1 mutations combined with resistance to ICB. The proposed rationale was that LKB1 loss silences the stimulator of interferon genes and finally promotes the expression of type 1 interferons and other cytokines, favoring an immunosuppressive TME, which contributes to ICB resistance [98]. DDRi increasing DNA damage can reset the inflammatory microenvironment of tumors to a higher level, thereby enhancing cytokine gene expression in vivo or in vitro (CCL2, CCL3, CCL5, and CXCL10). The inflammatory cytokine increase impacts the TME by recruiting immune cells and contributes by making it more immunogenic. In summary, regarding the immunologic landscape, alterations of DNA replication amplified by DDRi favor immunological vulnerabilities in tumors by attracting effector immune cells while contemporaneously inducing immunosuppressive pathways, thus providing the rationale for the combinatorial therapy of DDRi with ICB [98].

#### 1.2.3. Clinical Trials with ICIs and DDRi Combinatorial Therapy

In a first reported phase I clinical trial [105], 83% and 75% disease control rates were observed with the administration of durvalumab plus olaparib and durvalumab plus cediranib, respectively. However, the response to therapy was independent of PD-L1 expression, and grade 3–4 aEs occurred in a few patients. The phase Ib/II Topacio/Keynote-162 trial conducted with PARPi niraparib combined with pembrolizumab recruited 55 previously treated advanced or metastatic TNBC patients [106]. In 15 evaluable patients with tumor BRCA mutations, the ORR, disease control rate, and median PFS were 47%, 80%, and 8.3 months, respectively, while in another 27 evaluable patients with BRCA wild-type tumors, these values were 11%, 33%, and 2.1 months, respectively. The most common treatment-related AE of grade 3 or higher ranged from 7% (fatigue) to 18% (anemia); iraEs of grade 3 were observed in two patients (4%). The authors concluded that niraparib plus pembrolizumab results in a promising antitumor activity with higher response rates in patients with tumor BRCA mutations. The phase I/II MEDIOLA study that administered olaparib with durvalumab recruited four cohorts with different cancers. The breast cancer cohort included 34 patients with gBRCA1/2 or both mutations, progressive, HER2-negative, and metastatic breast cancer [107]. Patients had not received more than two previous lines of CT for metastatic breast cancer. Eleven (32%) patients experienced grade 3 or worse aEs and three (9%) patients discontinued due to aEs. A similar ORR was observed in 17 TNBC and 13 HR+ subjects (59% and 69%, respectively). The median PFS was longer in the 13 patients with HR+ disease (9.9 months and 4.9 months, respectively). Again, OS was similar in both subgroups (22.4 months and 20.5 months, respectively). Although a definitive answer to whether ICIs combined with DDRi improve the outcomes is not yet available, this combinatorial therapy is under investigation in the KEYLYNK-009 trial (NCT04191135) [108] also as maintenance treatment following induction CT independent of the mutational status.

### 1.3. Monoclonal Antibodies against Human Epidermal Growth Factor Receptor 2 (HER-2) Family in HER2+ Breast Cancer

HER2 overexpression identifies a breast cancer molecular subtype with worse outcome [8,109]. mAb targeting specific HER2 epitopes was the first anti-HER2 immunotherapy. They, upon binding, neutralize the target molecule expressed by cancer cells, thereby inhibiting their proliferation and survival [110]. Trastuzumab, a humanized IgG1 mAb, was the first HER2-specific mAb approved for HER2+ breast cancer in 1998 [111]. Trastuzumab, by blocking HER2 signaling, promotes G1 cell-cycle arrest and inhibits the PI3K pathway, favoring apoptosis and angiogenesis inhibition. First, trastuzumab, combined with first-line CT, obtained approval for metastatic HER2+ breast cancer [112]; then, after showing efficacy and safety in early breast cancer, it also became a conventional neo-adjuvant or adjuvant treatment in association with different anti-HER2 therapies and/or CT [9,30,113]. The rationale for combining anti-HER2 therapies with CT is their synergistic effect, which could likely be due to the depletion of DNA repair activity by the binding of antibodies to the epidermal growth factor receptor extracellular epitopes or to HER2 itself [114,115]. Following trastuzumab, specific antibodies targeting different HER2 epitopes and capable of inhibiting other cellular signaling pathways have been built. These mAbs could also elicit an indirect antitumor response on effector iCs by engaging with Fc receptors [68]. In fact, they can activate a complete (innate and adaptive) immune response inducing antibody-dependent cytotoxic cellular killing of HER2-overexpressing cells by NK cells; moreover, through the presentation of HER2 by MHC-I molecules, they can activate cytotoxic T lymphocytes and decrease intra-tumoral Tregs [116]. Pertuzumab is another IG1 mAb targeting the domain II of the HER2 receptor, thus preventing HER2 hetero-dimerization with HER3. It inhibits intracellular signaling via the PI3K/AKT pathway and can elicit antibody-dependent cellular cytotoxicity [113,117]. Pertuzumab has received approval for the treatment of metastatic and early-stages HER2+ breast cancer patients. Pertuzumab in combination with trastuzumab and CT improved OS in metastatic HER2+ breast cancer patients versus trastuzumab and CT alone [118,119,120,121]. Similarly, pertuzumab showed clinical benefit in the neo-adjuvant and adjuvant setting again in combination with trastuzumab and CT [122,123,124,125]. Overall, the use of trastuzumab and/or pertuzumab significantly increased the ORR and increased for a few months the PFS and OS [126]. However, due to intrinsic or acquired resistance, only roughly one-third of HER2+ tumors benefit from anti-HER2 antibodies [127]. To overcome the resistance to anti-HER2 antibodies, in addition to their combination with conventional CT, further strategies such as improving their functionality to enhance antibody-dependent cellular cytotoxicity have been applied. This has been attained through modification in the Fc region similar to margetuximab, a chimeric IgG1 anti-HER2 mAb [128], or with an antibody–drug conjugate (ADC) that covalently conjugates the tumor-specific mAb with a cytotoxin acting within the microtubules, thereby increasing antitumor immunity. This category includes trastuzumab-emtansine (T-DM1) [129,130], trastuzumab deruxtecan [131] and, more recently, sacituzumab-govitecan [132]. On the other hand, the higher PD-L1 expression in HER2+ breast cancer patients suggests a combination of anti-HER2 mAbs and ADCs with ICIs.

#### Anti-HER2 mAbs and/or ADC Combined with ICIs

Some trials have assessed ICIs in combination with trastuzumab or ADCs. The phase Ib/II PANACEA and phase Ib CCTG IND 229 trials take part in the former type of studies. In the PANACEA trial [65], PD-L1 positivity was defined as having a combined positivity score of >1, initially evaluated by QualTek and then by the Dako22C3 assay. The 58 women, who were recruited after they had progressed on the previously mentioned trastuzumab therapy, received pembrolizumab plus trastuzumab. Of these, 46 patients were PD-L1+ and had a 15% ORR and a disease control rate of 24% with no response or disease control observed in those who were PD-L1−. In the other trial, in which recruited patients had previously been treated with trastuzumab, pertuzumab, and a taxane, and were then given durvalumab and trastuzumab, no response was recorded [133]. In the phase II Kate2 trial [134], patients who were previously treated with trastuzumab and a taxane were randomized to receive atezolizumab or placebo associated with T-DM1. A trend toward a prolonged median PFS only was found in PD-L1 IC+ patients, with a PFS of 8.5 months in patients who were given the combination vs. a PFS of 6.8 months in those who received T-DM1 alone. Heavy aEs occurred more often in patients who were given the combination therapy including pyrexia and grade 3 transaminitis in 35% and 8% of patients, respectively, and one death. A similar cohort was recruited in a phase Ib trial where the 20 evaluated patients received pembrolizumab plus T-DM1 [135]. For this study, the ORR was 20% and the median PFS was 9.6 months; a trend toward higher response occurred in patients with a PD-L1 combined positivity score of <1 and TILs ≤ 10%; furthermore, 10% of the patients complained of grade 3 transaminitis. Other trials investigating NACT combined with ICIs and anti-HER2 therapies are ongoing. In the Impassion 050 trial, patients were randomized to receive atezolizumab or placebo concomitant with NACT, including doxorubiicin + cyclophosphamide, followed by paclitaxel, trastuzumab, and pertuzumab [133,136]. Atezolizumab or placebo was continued in an adjuvant setting together with trastuzumab and pertuzumab for 52 weeks overall. However, the study was stopped due to an unfavorable risk–benefit ratio.

### 1.4. Other Investigational Immune-therapies

Adoptive cell therapy (ACT) is an immunotherapy strategy that was started in 1987 and has since then evolved in different forms. First, it was based on the autologous or allogenic transfer of TILs following isolation from TME, ex vivo activation, and expansion through the use of interleukin-2, and infusion back to the patient. However, multiple obstacles, the most likely among them being an immunosuppressive TME, impeded this technique to exit from the investigational ground and spread out [137,138]. More recently, a similar approach using NK cells, unlike TILs, has been applied to a small sample size of breast cancer patients. This trial was based on the rationale that NK cells benefit from the ability to kill cancer cells in an MHC-independent and non-tumor antigen-restricted way. However, the outcome was unsatisfactory, possibly because of the same drawbacks that occurred with TILs [139]. Therefore, two different modalities were developed, the use of T cells that have been genetically engineered to express modified T-cell receptors or chimeric antigen receptors (CAR). In the former technique, the alpha-beta TCRs are armed with synthetic T cell receptors for targeting cancer-specific epitopes presented by MHC molecules. This procedure enhances T cell affinity toward the cancer cells presenting the targeted epitope, although the low number of potential targets is the main limitation. Clinical trials with intravenous infusion of TCR-modified T cells against HER2, MAGE-A3, or other antigens are ongoing [138]. Recently, gamma delta T cells have been considered for ACT due to their ability to recognize and kill cancer cells in a human leucocyte antigen (HLA)-independent manner, together with the ability to induce antibody-dependent cellular cytotoxicity. In addition, these T cells, such as NK cells, express activating receptors that bind their ligands present on the cancer cells. Two trials in this regard have been carried out with promising findings [140,141]. Further, CAR is a synthetic molecule that consists of an extracellular domain (scFv) combined with the intracellular signaling domain of a physiological T cell receptor (CD3 zeta chain), as well as different co-stimulatory domains (CD28, ICOS, and OX40). T or NK cells engineered to express CAR can identify multiple tumor cell surface antigens by their extracellular domain in a non-MHC-restricted way that can allow overcoming immune evasion due to MHC down-regulation. A few trials evaluating CAR-T therapy in breast cancer are ongoing [138,142]. In 1992, we recruited metastatic ER+ breast cancer patients with clinical benefit during the first-line anti-estrogens treatment. These patients additionally received an active immune stimulation using the interferon-beta interleukin-2 sequence. The promising results of the pilot study with this active immunotherapy were first reported in 2005 due to the low accrual rate in our oncological center [143,144]. Subsequently, the unexpected difficulties we encountered in launching a sponsored prospective confirmatory randomized clinical trial convinced us to resort to a more feasible 2:1 control-case retrospective observational study that included 95 controls treated with hormone therapy alone and 42 cases treated with hormone therapy plus the interferon-beta-interleukin-2 sequence. Twenty-eight controls (28.9%) unlike cases had received biological drugs including cyclin kinase inhibitors (CKIs) and most cases were given first-line SERMs or SERDs unlike controls who were treated with first-line AIs. Despite this, the median PFS and OS were significantly longer in the 42 cases who had received additional immune-stimulating immunotherapy than in the 95 controls (Table 3). No grade 3–4 AEs occurred in the 42 cases; moreover, this proposed immunotherapy is 8–18 times cheaper than CKIs [145]. Table 3 shows some major recent clinical trials with anti-HER2 mABs, ADC, anti-HER2 mAb, or ADC plus ICIs or other investigational immunotherapies.

### 1.5. The Tumor Immunogenicity: A True or an Incomplete Understanding?

In the scientific literature, the host immune system’s capability to provide a spontaneous immune response to cancer cells is a well proven and consolidated concept [146]. The intensity of this response defines tumor immunogenicity. More recent investigations support the notion that the TILs rate in the TME, tumor mutation burden, and neo-antigens, as well as other genetic alterations such as MMRd, microsatellite instability, HRD and defects in DDR genes, are tools for tumor immunogenicity assessment [147,148] and potential surrogate biomarkers that are predictive of a better immune response. Based on this, breast cancer is considered a low or moderately immunogenic tumor, unlike other types of cancers such as melanoma and lung cancer, which are reported as highly immunogenic [14,71].

#### 1.5.1. The Immunological Balance Resulting from the Interplay between Cancer Cells and TME: Mechanisms Inhibiting or Promoting an Antitumor Immune Response

The TME contains many different cells and immune-modulatory humoral factors that, overall, contribute to the immune balance. Cellular components include stromal cells, together with cells of the innate and adaptive immune system. Cytokines, chemokines, and growth factors that are secreted by cancer, stromal, and immune cells are the immune-modulatory humoral factors, which, together with tumor-cell-derived exosomes, play a central role in the cell-to-cell communication and the crosstalk between cancer cells and their TME. The commonly prevailing immunosuppressive TME results from the balance of opposite actions that, directly or indirectly, induce or inhibit the cell-mediated immune response and tumor growth. Due to the widely documented spatial intra-tumor and inter-tumor genetic heterogeneities [149], the immunological balance differs in different tumors of the same type and different organs of the same tumor. Additionally, temporal evolution occurs after therapeutic interventions, together with genetic and metabolic alterations. This makes immunological balance a dynamic condition that changes over time [150,151].

#### 1.5.2. Anti-Tumoral Immune Activities

TILs that can be present in breast cancer TME are made of cytotoxic (CD8+) cells, helper (CD4+) T cells, Tregs, and NK cells [152]. A higher TILs rate in TME has been reported to be associated with improved clinical outcomes in both HER2+ and TNBC [153]. Additionally, in HER2+ breast cancer, a higher TILs rate in TME directly correlates with pCR to NACT. Therefore, overall, despite the presence of immunosuppressive Tregs, TILs are commonly thought to reflect a state of immune activation. Regarding B cells, findings suggest that their principal role is to support an immunological response by producing antibodies and inducing an optimal T cell activation and cellular immunity [154]. NK cells, also defined as CD3− CD56+ cells, comprehend two subsets, CD56^bright^ CD16^low/−^ and CD56^bright^ CD16^+^. The former can recognize cancer cells in a nonrestricted MHC class I modality and kill them by releasing cytolytic granules containing perforins and granzymes [17]. Activated dendritic cells (DCs) play a key role in the immunological response through the presentation or cross-presentation of tumor antigens to the CD4+ and CD8+ T cells, which is essential for the maturation and activation of tumor-specific cytotoxic T lymphocytes that move to the tumor’s niche to eliminate tumor cells. Among tumor-associated macrophages (TAMs), the M1-like phenotype typically performs anti-tumor functions and is capable of directly killing tumor cells through the release of reactive oxygen species and nitric oxide, and antibody-dependent cellular cytotoxicity [155]. Among tumor-associated neutrophils (TANs), the N1 phenotype induces inflammation processes through the release of reactive oxygen species as well as interleukin (IL)-1beta, TNF-alpha, IL-6, and IL-12 cytokines [156].

#### 1.5.3. Immune Activities Promoting Immune Inhibition and Tumor Progression

In the TME, Tregs, cancer-associated fibroblasts (CAFs), TAMs, myeloid-derived suppressor cells (MDSCs), and TANs are the principal cells that, together with many different humoral factors such as cytokines (IL-4, IL6, IL-10, IL-13, TGF-beta, and TNF alpha), chemokines, their ligands (CXCL12 and CXCR4), enzymes (such as indoleamine 2,3 dioxygenase (IDO) and ARG1), exosomes, and other factors (VEGF and prostaglandin E2), inhibit the immune response. The humoral factors are largely secreted by tumor cells and the other cells within the TME [149,157]. In TME, VEGF induces the proliferation of immunosuppressive cells, decreases T cell recruitment, and increases T cell exhaustion, in addition to stimulating tumor vessel growth [158].

##### Tregs, TAMs, and MDSCs

Tregs move to the TME recalled by the cytokines and chemokines secreted by tumor and immunosuppressive cells. In metastatic breast cancer, they can be derived by converting resting CD4+ T cells after induction by IL-10 and TGF-beta [159]. Tregs strongly promote immune inhibition and tumor progression by impairing the cytotoxicity of effector T cells, modulating antigen-presenting cells and inducing metabolism alterations [152]. In breast cancer, an increased ratio of total FoxP3+ Tregs to CD8+ cytotoxic T lymphocytes is an unfavorable prognostic index unlike a high CD8+ rate [160]. CAFs can secrete mitogenic growth factors, pro-angiogenic factors, and TGF-beta, overall favoring tumor progression [149]. In TME, TAMs are among the most abundant cell type found [161] and the M2-like phenotype inhibits cytotoxic T lymphocytes through the depletion of metabolites, secretion of cytokines and chemokines, and the expression of receptors/ligands for ICs [138]. TAMs induce angiogenesis and tumor progression through the production of VEGF, other cytokines (IL-10, CCL2, CCL17, CCL22, and TGF-beta), and matrix-degrading enzymes [138]. TAMs attract Tregs by secreting chemokines [162] and inhibit cytotoxic T cells through an increase in interferon-gamma-induced PD-L1 expression [163]. MDSCs encompass a heterogeneous population of immature myeloid cells that favor immune suppression and tumor progression. The following mechanisms have been reported to induce immune suppression: (1) inhibition of T cell proliferation and promotion of apoptosis due to increased expression of IDO, which is an enzyme that accounts for tryptophan catabolism and kynurenine production [164]; (2) STAT3 hyper-activation correlating with a noncanonical NF-kB pathway, which leads to IDO up-regulation [165]; (3) secretion of IL-6, IL-10, and TGF beta, which are pro-inflammatory immunosuppressive cytokines [166]; (4) production of reactive oxygen and nitrogen species by attracting other immunosuppressive cells [167].

##### N2 Phenotype, CD56^bright^ CD16^+^ NK Subset, DCs, and Exosomes

The N2 phenotype of TANs exerts an immunosuppressive action by reducing the anti-tumor response of the CD4+ and CD8+ T lymphocytes [168]. In the TME, TGF-beta induces the formation of an N2 phenotype that inhibits T cell action through the enhanced production of inducible nitric oxide synthase and arginase [169,170]. Following the degradation of the extracellular matrix, TANs contribute to the release of VEGF, suppress NK-mediated tumor cell clearance, and increase the extravasation of cancer cells [171]. In ER+HER2− breast cancer, an increased neutrophils-to-lymphocytes ratio is directly correlated with a worse prognosis [172,173]. The CD56^bright^ CD16^+^ NK subset, unlike the CD56^bright^ CD16^low/−^ subset, can promote tumor progression through the matrix metalloproteinase 9, VEGF, and angiogenin secretion and release [174,175]. Interestingly, to highlight the dual role of NK cells, it has also been found that in ER+ and HER2+ breast cancer patients, the infiltration of NK cells correlated with an improved outcome, unlike in TNBC, where it combined with a worse prognosis [176]. A few tumor-infiltrating DCs with immature phenotypes promote endothelial cell migration and tumor growth through the production of pro-angiogenic factors, low expression of co-stimulatory molecules, and over-expression of regulatory molecules [177]. A DCs subset, termed plasmacytoid DCs, together with a decreased antitumor immune response, Tregs increase, and a higher rate occurred in TNBC compared to the other breast cancer subtypes [178]. Breast cancer cells as well as CAFs [179] and bone-marrow-derived cells [180] secrete exosomes that carry PD-L1 [181]. In the TME, exosomes reportedly act as vehicles that transport PD-L1 to different cell types, thereby regulating immune surveillance [182] and circulating exosomes from primary breast tumors recruit MDSCs to pre-metastatic sites [183]. Finally, a less efficient or decreased surface antigen expression of MHC in tumor cells and the up-regulation of immune checkpoint receptors are two main mechanisms of tumor immune evasion that are at least in part due to genetic instability inherent in all tumor cells together with the immune selection process [184]. This means that tumor immunogenicity as a result of genomic aberrations and the TILs rate is a large restraint of the immunological balance in the TME. In fact, genomic aberrations and TILs are only some of the multiple mechanisms contributing to the immune response that are counteracted by multiple others favoring immune inhibition (Figure 1). 

#### 1.5.4. Conditions Favoring a Successful TME Immune Manipulation

Thus far, it is unclear if PD-L1 expression is a predictive biomarker for the response to CIs [43,44]. Similarly, findings on genomic aberrations as predictive biomarkers for the response to DDRi or DDRi + ICB are limited and still need extensive validation in large and well-designed clinical trials [98]. Further, HER2+ is a biomarker predictive of the benefit arising from anti-HER2+ mAb in only one-third of HER2+ breast cancer patients [127]. Conversely, in most cases of ER+ breast cancer, ER expression is a highly predictive biomarker for a response or clinical benefit to anti-estrogens. However, ICIs, DDRi, and anti-HER2+ mAbs therapies affect one or a few pathological molecular pathways, likely without a significant impact on the immunosuppressive TME, while the ER-mediated anti-estrogen action in ER+ breast cancer involves multiple genes and several pathological pathways [157,185,186] favoring the reversion of the immunosuppressive TME. Accordingly, due to its potential capability to revert the immunosuppressive TME, anti-estrogen therapy has been proposed in ER+ cancers, apart from breast cancer, as a treatment that synergizes with conventional therapy [187]. Moreover, ICIs, DDRi, and anti-HER2+ mAb, unlike anti-estrogens, are given in combination with conventional CT, as they often do not work when given alone. In our observational study [145], despite hormone-receptor-positive breast cancer being considered a less immunogenic molecular subtype, anti-estrogens boosted tumor immune response. We hypothesized that in our studied patients, the prolonged G0-G1 state, often concomitant with a decreased tumor burden, accounted for a down-regulation of the main hallmarks sustaining tumor growth, including immune evasion with a large reversion of the immunosuppressive TME [145,188]. A significant peripheral blood increase in T and NK effector cells occurred in all cases that received additional immune-stimulating therapy [189]. In two other pilot studies in which metastatic breast cancer patients had received maintenance-immune-modulating therapy consisting of interleukin-2 and retinoic acid or interferon-β and retinyl palmitate, the immune-modulating maintenance treatment and a concomitant low tumor burden likely favored an immune response and accounted for the significantly improved clinical outcome [190]. Consistently, it has been recently stated that “there are some possible biological explanations to justify why immunotherapy can benefit TNBC patients regardless of PD-L1 positivity in localized disease, but not in the metastatic setting. The host immune system is probably more robust in the early disease due to the limited cancer burden and the major effectiveness in triggering an antitumor immunologic response to new antigens” [43]. We have also recently reported on one breast cancer patient with minimal residual disease (MRD) at high risk of relapse following radical resection of a single lung metastasis and two others with biochemical recurrence after radical prostatectomy who received successful immune therapy [191]. In conclusion, in addition to a prolonged G0-G1 state, our findings along with others suggest a low or undetectable tumor burden for more successful tumor immune manipulation [157].

#### 1.5.5. Metastatic Breast Cancer as an Incurable Disease

Overall, about 20% of breast cancer patients experience recurrence or metastasis within the first five years after primary radical surgery [192]. Recently, rates of distant metastases, ranging from 23.9% in luminal A (ER+ or PR+ and HER2−) to 32% in basal (ER− or PR− and HER2−) and 52.4% in HER2+ (ER− or PR− and HER2+), have been reported in a study conducted in 324 eligible patients with a median follow-up of 7.3 years [193]. In metastatic breast cancer, independent of molecular subtype, the five-year cancer-specific survival rate is approximately 29%, which reduces to 12% in the case of metastatic TNBC [194]. Although two decades have passed since precision medicine and targeted therapies usually combined with conventional chemo or hormone therapy started in oncology, the metastatic disease clinical outcome has improved in small subsets [127,138,195]. The median PFS and/or OS have increased by a few months or a few years at best, and death inevitably occurs in most patients [138,195]. Moreover, due to these targeted treatments, a significant percentage of patients suffer from additional heavy side-effects including irAEs [196,197,198]. Overall, this and the advances in the specific biology of cancer growth and progression suggest a novel therapeutic strategy aimed at obtaining a definite cure by preventing overt metastatic disease.

### 1.6. Locally Advanced Breast Cancer Patients: A New Paradigm for Additional Adjuvant Therapy

#### 1.6.1. The Conventional Adjuvant CT and the Formation of the Pre-Metastatic Niches (PMNs)

In the Gompertz mathematical model first proposed by Goldie and Coldman in 1979 [199,200], the probability of the existence of nonresistant cells exponentially decreases as the size of the tumor grows, and the double time of cancer cells only begins to increase as the tumor size becomes larger. Eventually, the proliferation rate slows down as the cancer population asymptotically approaches the plateau population; then, the growth process stops due to the lack of space and oxygen. The Gompertz function realistically represents tumor growth because it limits the population growth as the tumor size attains the carrying capacity of the host organ. Since then, the Gompertz model of tumor growth by Goldie and Coldman has been elaborated upon several times [201,202]. This model remains the milestone on which adjuvant CT and radiotherapy (RT) schedules have been designed both regarding their administration timing and dosage and, in the case of CT, for the type of drugs to be combined. Consistent with this mathematical model, the first 4–6 months following the radical resection of a primary tumor is considered the optimal interval time for cyclic adjuvant CT administration. Despite this, large scientific documentation in the last two decades suggests the dissemination of cancer cells at in situ stages well before the detection of a primary tumor [190]. Furthermore, the heterogeneity of “early” metastases in breast cancer occurs due to different microenvironments [203], also termed pre-metastatic niches (PMNs) [204,205]. The formation of PMN involves three successive temporal phases: (a) first, the metastatic microenvironment is educated by the primary tumor; (b) second, the secondary sites recruit immunosuppressive cells; (c) finally, the circulating tumor cells move to PMN [206,207,208]. The process, termed “organo-tropism” or cancer cell “homing,” is driven by specific gene expression and chemokine secretion. However, an increasingly relevant role in the formation of the PMN is attributed to exosomes. Extracellular vesicles and cancer-cell-derived exosomes can interact with inflammatory cytokines (IL-6 and IL-8) to promote the formation of PMN [209,210]. Exosomes promote angiogenesis and permeability in PMN through micro-RNAs secretion [211,212,213] and by carrying many pro-angiogenic molecules including VEGF and matrix metalloproteinases [214,215]. In addition, exosomes are also involved in stromal PMN remodeling by triggering the differentiation of cells [216] or normal breast cancer fibroblasts [217] to CAFs or pro-metastatic CAFs [218].

#### 1.6.2. The Specific Biology of Disseminated Cancer Cells (DCCs)

“Early” DCCs, also named MRD, evolve and acquire specific characteristics that allow them to generate micro-metastases resistant to therapies focused on eliminating the tumor burden residual to the surgical resection of the primary tumor. A population-level dormancy and cellular dormancy models that are not mutually exclusive and co-evolve with the maturation of the metastatic niche have been described in the study. In the former model, limited blood supply, secondary immune-editing, or apoptosis inhibit the expansion of proliferating DCCs, while in the latter model, cells enter the G0-G1 state. Thus, in the “host” metastatic microenvironment of solid cancers, DCCs may enter into a quiescent state to “awaken” and lead to clinically/radiological detectable metastases after years or even decades. During this prolonged interval time, they interact with many micro-environmental signals that can trigger the proliferation process. Therefore, the Gompertz mathematical model of tumor growth proposed by Goldie and Coldman does not fit the specific biology of DCCs that, recently, we have widely reported on [190]. Epigenetic reprogramming mechanisms triggered by micro-environmental or intra-tumor signals likely induce the long-term commitment of DCCs to quiescence while retaining growth potential [219]. We have represented this condition at the PMN as the counterbalancing of immunological and nonimmunological micro-environmental or intra-tumor signals governing an unstable virtual equilibrium line [190]. This instability over time exerts proliferative foci that progress or not depending on the environmental status of the involved niche at that specific time. Therefore, the principal aim should be to turn for a prolonged time, possibly indefinitely, the unstable to a more stable virtual equilibrium line while switching off the proliferative foci [190]. Figure 2 shows in detail this proposal for cases of breast cancer at a high risk of relapse based on additional cyclic immunotherapy combined or alternated with prolonged conventional adjuvant hormone therapy or CT, respectively. We carried out these or similar schedules for 5–6 years or until relapse with preliminary promising findings. Moreover, while the short cyclic CT administration makes it well tolerated [220], the proposed immune therapies did not show relevant AEs and irAEs [145,191,220].

## 2. Discussion and Conclusions

Since a long time in the modern era, oncologists have been fascinated by the possibility of using the immune system of a cancer patient to fight cancer. This attractive hypothesis would have allowed cancer to be cured without relevant toxicity. The efforts to pursue this aim enormously increased recently when it was unequivocally proven that, from the beginning, the host immune system can recognize and activate a cell-mediated innate and adaptive immune response against tumor cells [221]. Breast cancer is the most common form of cancer and the second cause of death by cancer in women. In 2020, it represented 30% of female cancers with 276,480 new cases and more than 42,000 estimated deaths [222]. Although breast cancer is considered less immunogenic than others, the HER2+ and TNBC subtypes are thought to be more immunogenic due to a higher tissue TILs rate, genetic alterations, and/or PD-L1 over-expression. Nevertheless, in both these subtypes, a minority of patients benefit from the active immunotherapy given directly with anti HER2+ mAbs or indirectly by ICIs, usually combined with conventional hormone therapy or CT, with concomitant grade 3–4 AEs in about 10–20% of patients. The propensity to induce an immune response is likely inadequately evaluated by a high TILs or CD8+ rate in TME, tumor mutational burden, and other genetic aberrations in tumor cells. In fact, at the TME site, genetic aberrations and TILs or CD8+ are some of the factors contributing to the immune response among the multiple known and unknown mechanisms contributing to the immune balance. Additionally, the extent of any single contribution to the immune balance is unknown. Thus, any cancer type, independent of the currently pre-defined immunogenicity, has inducible immunogenicity proportional to the capability of the available therapeutic means to counteract tumor growth and immune inhibition. The surprisingly promising findings observed in our 2:1 control-case retrospective observational study support this notion; this, along with other findings, suggests that low tumor burden and/or tumor cell quiescence are important conditions favoring a successful immune manipulation [190,191]. This assumption and the absence of therapies to cure the overt metastatic disease suggest concentrating any effort during MRD where both conditions coexist. Our proposed strategy combines a prolonged cyclic immune manipulation with conventional hormone or short-term CT to indefinitely maintain the PMN tumor cell quiescence while switching off the occasionally proliferating foci. This therapeutic strategy could significantly decrease the rate of relapses in the first 5–6 years with a huge saving of expenses by the Health National Services and, more importantly, of lives and pains. Based on our and others’ preliminary work and pilot studies [145,191,220], specific clinical trials should be easy to design, and safe and cheap to carry out. Multinational drug companies do not have an interest to sponsor such investigational trials as the proposed schedules include repurposed drugs without patents and at low cost. Despite this, one can expect that governmental drug agencies or private institutions will start such trials that, if successful, can be extended to other solid tumors.

## Figures and Tables

**Figure 1 biomedicines-10-02511-f001:**
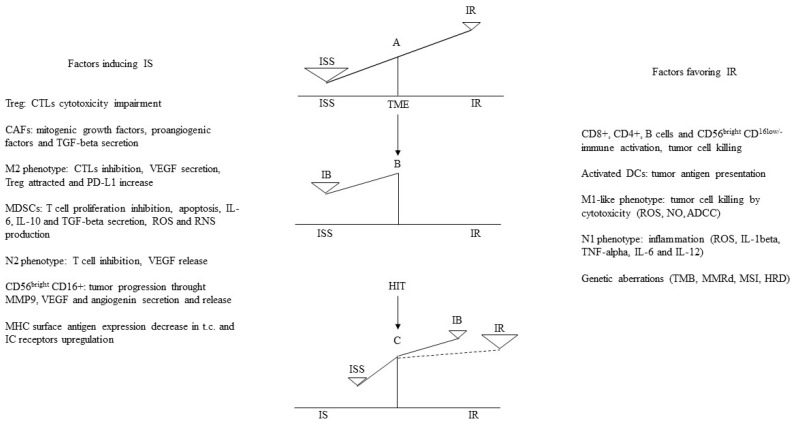
A-B: In TME, factors inducing ISS commonly by far overcome those favoring IR, thus with regard IB, ISS prevails. C: In ER+ metastatic breast cancer patients in clinical benefit during hormone immunotherapy (HIT), SERMs combined with interferon-beta-interleukin-2 sequence (INF-beta-IL-2) counteract tumor proliferation and immune inhibition and stimulate significant immune response [145], therefore IR prevails in TME. ISS: immune suppression; IR: immune response; IB: immune balance; TME: tumor microenvironment; TILs: tumor infiltrating lymphocytes; DCs: dendritic cells; M: macrophage; N: neutrophil; ADCC: antibody-dependent cell-mediated cytotoxicity; Treg: regulatory T cells; CAF: cancer-associated fibroblast; TGF-beta: transforming growth factor beta; CTLs: cytotoxic T cells; VEGF: vascular endothelial growth factor; PD-L1: programmed death ligand 1; MDSC: myeloid derived suppressor cell; IL: interleukin; ROS: reactive oxygen species; NO: nitric oxide; RNS: reactive nitrogen species; MMP-9: matrix metallo proteinase protein 9; MHC: major histocompatibility complex-1; IC: immune checkpoint; TMB: tumor mutation burden; MMRd: mismatch repair deficiency; MSI: microsatellite instability; HRD: homologous recombination deficiency.

**Figure 2 biomedicines-10-02511-f002:**
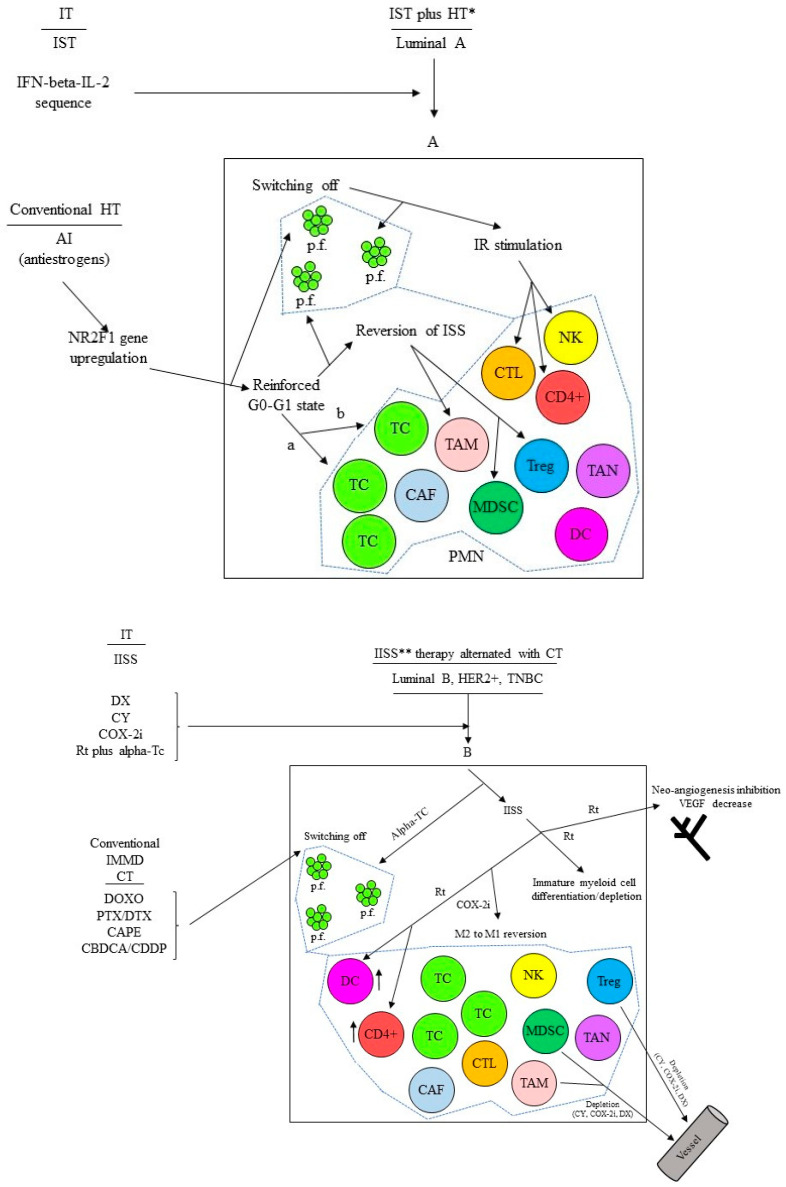
(**A**) Additional cyclic immunotherapy (IT) concomitant with prolonged conventional hormone therapy (HT) in luminal A breast cancer at high risk of relapse. * For the schedule see references [143,145]. IST: immunostimulating therapy; IR: immune response; ISS: immune suppression; IFN: interferon; IL-2: interleukin-2; AI: aromatase inhibitor; a: population-level dormancy model; b: cellular dormancy model; PMN: pre-metastatic niche; p.f.: proliferation focus; PMN borders. (**B**) Additional cyclic immunotherapy (IT) alternated with prolonged conventional adjuvant chemotherapy (CT) in breast cancer other than luminal A and at high risk of relapse. IISS: inhibiting immune suppression; IMMD CT: immune modulatory CT. ** For the schedule see reference [191], this IISS therapy is alternated with four cycles (3 months overall) of conventional IMMD CT; DX: desamethasone; CY: cyclophosphamide; COX-2i: COX-2 inhibitor; Rt: retinyl palmitate; alpha-Tc: alpha tocopheryl acetate; DOXO: doxorubicin; PTX/DX: paclitaxel/docetaxel; CAPE: capecitabine; CBDCA/CDDP: carboplatin/cisplatin; PMN borders.

**Table 3 biomedicines-10-02511-t003:** Major clinical trials recently conducted with anti-HER2 monoclonal antibodies (mAbs), antibody–drug conjugates (ADCs), their combination ICIs, and other investigational immunotherapies.

Trial	Phase	Drug	Setting	Treatment	Pts (n)	Outcome	Ref.
*Anti HER2 mAbs*							
Cleopatra NCT00567190	III	Pertuzumab (PERT)- Trastuzumab (TRST)	HER2+, l. a. or mBC	TRST + DCX + PB vs. TRST + PERT + DCX	406 vs. 402	mOS 40.8 vs. 56.5 mo. mPFS 12.4 vs. 18.7 mo.	[121]
Berenice NCT02132949e	II	PERT-TRST	HER2+, localized BC, neoadjuvant	DOXO + CY followed by PTX + TRST + PERT vs. FEC followed by DCX + TRST + PERT	109 vs. 198	pCR 61.8% vs. 60.7% 6.5% vs. 2% pts with at least one LVEF decline	[124]
*Antibody–drug conjugates (ADCs)*							
Kamilla NCT01702571	IIIb	Ado-trastuzumab emtansine (T-DM1)	Pre-treated, l.a. or mBC with brain metastases (BM)	T-DM1	398	ORR 21.4%, CBR 42.9% (in 126 pts with measurable BM) PFS 55 mo., OS 18.9 mo.	[129]
Th3resa NCT01419197	III	T-DM1 vs. physician choice	Pre-treated, HER2+ aBC	T-DM1 vs. physician choice	404 vs. 198	mOS 22.7 vs. 15.8 mo. Serious AEs 25% vs. 22%	[130]
Destiny-Breast 01	II	TRST-Deruxtecan	Pre-treated, HER2+ mBC	TRST-Deruxtecan, 3 different doses	184	mRD 14.8 mo., mPFS 16.4 mo. with the recommended dose Grade 3–4 AEs ranging from 7.6 to 20.7%	[131]
IMMU-132-01	I/II	Sacituzumab-govitecan hziy (Trop2 + SN38)	Pre-treated mBC	Sacituzumab-govitecan hziy	108	ORR 34.3%, MRD 9.1 mo., CBR 45.4%, mPFS 5.5 mo., mOS 13 mo.	[132]
*Anti HER2 mAbs or ADCs plus ICIs*							
PANACEA	Ib/II	Pembrolizumab (PE)-TRST	HER2+ BC progressing on prior TRST	TRST + PE	58	ORR 15%, DCR 24% in PDL1 + vs. no ORR in PDL− pts	[64]
KATE2	II	Atezolizumab (ATZ)-T-DM1	Pre-treated, HER2+ aBC	ATZ + T-DM1 vs. PB + T-DM1	133 vs. 69	mPFS 8.5 vs. 6.8 mo. in PDL1+	[134]
*Other investigational immunotherapies*							
SOPHIA NCT02492711	III	Margetuximab (MAR) (chimeric antigen receptor)	Pre-treated, HER2+ mBC	MAR + CT vs. TRST + CT	266 vs. 270	ORR 22% vs. 16% mPFS 5.7 mo. vs. 4.4 mo. mOS 21.6 mo. vs. 19.8 mo.	[142]
2:1 control-case observational study	II	Sequential IFN-beta-IL-2 plus SERMs	Forst line ER+, HER2− mBC	Sequential IFN-beta-IL-2 plus SERMs vs. AI	42 vs. 95	mPFS 33 mo vs. 18 mo. mOS 81 mo. vs. 62 mo.	[145]

mOS: median overall survival; mPFS: median progression-free survival; pCR: pathological complete response; LVEF: left ventricular ejection fraction; DCR: disease control rate; CBR: clinical benefit rate; MRD: median response duration; AEs: adverse events; Trop2: humanized mAB targeting the human trophoblast cell-surface antigen; SN38: active metabolite of irinotecan (a topoisomerase inhibitor); PB: placebo.

## Data Availability

Not applicable.

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
