# Peer review of "Immune Checkpoint Inhibitors and Other Immune Therapies in Breast Cancer: A New Paradigm for Prolonged Adjuvant Immunotherapy"

_biomedicines, 2022, doi:10.3390/biomedicines10102511_

Round 1

Reviewer 1 Report

In the present review article, the authors have gathered information about the importance of immune checkpoint inhibitors in breast cancer. This is an interesting and updated review and could be a good source of information for a wide range of readers including molecular biologists, oncologists, and cancer biologists, however, there are a few points that need to be addressed by the authors:

1-The title is too long and could be shortened.

2-The manuscript has multiple typos and grammatical mistakes that need to be removed.

3-Some paragraphs are way too long and could be divided into multiple and relevant paragraphs to make it easier for the readers to follow and apprehend.

4-The authors could avoid unnecessary abbreviations throughout the manuscript.

Author Response

1-The title is too long and could be shortened.

The title has been shortened.

2-The manuscript has multiple typos and grammatical mistakes that need to be removed.

The revised version underwent a careful professional grammar and language editing.

3-Some paragraphs are way too long and could be divided into multiple and relevant paragraphs to make it easier for the readers to follow and apprehend.

The longest paragraphs, namely paragraphs 1.1.2., 1.5.3. and 1.6.1. have been divided and in the revised version each of them includes 2 subheadings.

4-The authors could avoid unnecessary abbreviations throughout the manuscript.

Useless abbreviations have been removed all over the text.

Reviewer 2 Report

I have reviewed the submission titled "Immune checkpoint inhibitors and other immune-therapies in breast cancer: tumor immunogenicity and a new paradigm for additional prolonged adjuvant immunotherapy combined with conventional hormone or chemotherapy" and recommend publishing the work. 

The authors have provided a comprehensive review of available immunotherapy-based options for breast cancer patients. The manuscript thoroughly reviews the application of check-point inhibitors as monotherapies and in combination with other agents. Alternate therapies, including investigational options such as mAbs and ADCs, are well described. The tables are very helpful for the reader as they summarize the clinical outcomes along with the therapeutic entity. Lastly, the work is concluded with adequate observations about using mAbs and combination hormone therapy. The authors also provide proposed strategies for future studies and clinical trial designs. 

Author Response

The authors thank the reviewer for its evaluation